# Lateral Root and Nodule Transcriptomes of Soybean

**Sajag Adhikari [1,2], Suresh Damodaran [1,3] and Senthil Subramanian [1,4,*]**

[1]  Department of Agronomy, Horticulture and Plant Science, South Dakota State University,
     Brookings, SD 57007, USA; sadhikari2@unl.edu (S.A.); sureshd@wustl.edu (S.D.)
[2]  Department of Agronomy and Horticulture, University of Nebraska, Lincoln, NE 68583, USA
[3]  Department of Biology, Washington University in St Louis, St Louis, MO 63130, USA
[4]  Department of Biology & Microbiology, South Dakota State University, Brookings, SD 57007, USA
[*]  Correspondence: Senthil.Subramanian@sdstate.edu; Tel.: +1-605-688-5623

**Abstract:** Symbiotic legume nodules and lateral roots arise away from the root meristem via dedifferentiation events. While these organs share some morphological and developmental similarities, whether legume nodules are modified lateral roots is an open question. We dissected emerging nodules, mature nodules, emerging lateral roots and young lateral roots, and constructed strand-specific RNA sequencing (RNAseq) libraries using polyA-enriched RNA preparations. Root sections above and below these organs, devoid of any lateral organs, were used to construct respective control tissue libraries. High sequence quality, predominant mapping to coding sequences, and consistency between replicates indicated that the RNAseq libraries were of a very high quality. We identified genes enriched in emerging nodules, mature nodules, emerging lateral roots and young lateral roots in soybean by comparing global gene expression profiles between each of these organs and adjacent root segments. Potential uses for this high quality transcriptome data set include generation of global gene regulatory networks to identify key regulators; metabolic pathway analyses and comparative analyses of key gene families to discover organ-specific biological processes; and identification of organ-specific alternate spliced transcripts. When combined with other similar datasets, especially from leguminous plants, these analyses can help answer questions on the evolutionary origins of root nodules and relationships between the development of different plant lateral organs.

**Dataset:** The dataset has been deposited to NCBI Gene Expression Omnibus (GEO) and has been assigned the accession number GSE129509. Raw data files have been deposited to NCBI's Sequence Read Archive (SRA) and can be accessed via links available at the GEO record URL: https://www.ncbi.nlm.nih.gov/geo/query/acc.cgi?acc=GSE129509.

**Keywords:** *Bradyrhizobium diazoefficiens*; RNA Sequencing; root lateral organ; legume

## 1. Summary

Root lateral organs, such as lateral roots (LRs) and symbiotic nodules, are unique in that they arise from differentiated cells, whereas shoot lateral organs arise from founder cells in the meristem [1,2]. LRs are present in all higher plants and are initiated in response to both developmental and—primarily—abiotic environmental cues (e.g., nutrients, mechanical stimuli). Along with LRs, roots of some higher plants are capable of forming nodules in association with nitrogen-fixing bacteria, termed rhizobia and *Frankia*.

There are two major types of nodules formed in legume roots: Indeterminate and determinate (reviewed in [3,4]). Indeterminate nodules are oblong and characterized by the presence of a persistent

nodule meristem, analogous to LRs. Examples of plants that form indeterminate nodules include temperate legumes, viz. *Pisum sativum* (pea), *M. truncatula* (Barrel Medic) and *Trifolium* species (clover). In contrast, determinate nodules are spherical and lack a persistent nodule meristem. There is no sustained cell division during determinate nodule development; nodule growth is more a result of cell expansion rather than cell division. Examples of plants producing determinate nodules include tropical/subtropical legumes, viz. *Glycine max* (soybean), *Vicia faba* (common bean), and *Lotus japonicus*. Additionally, indeterminate nodules arise from inner cortical cell layers, whereas determinate nodules arise from outer cortical cell layers.

Nodules appear to share developmental pathways with LRs, suggesting a root evolutionary origin for nodules [5]. Indeed, nodules appear to possess root identity as loss of function of *BLADE-ON-PETIOLE* orthologs in both *Medicago truncatula* and pea, leading to the development of roots from nodule vascular initials [6]. However, a legume nodule differs from the root in both development and morphology [4,7]. LRs arise from a few initial pericycle cells adjacent to a xylem pole and undergo a defined program of cell division and expansion. For example, in *Arabidopsis thaliana*, the formation of LRs occurs by a coordinated division of the pericycle cells and can be divided into eight stages based on anatomical characteristics and cell divisions [8]. In *Medicago truncatula*, LR formation involves cell division in the pericycle, as well as the endodermis [9]. Root nodule initiation also involves a defined program of cell division and expansion, but the site of initiation of legume nodules is the cortex cells (with a few exceptions [10,11]). Furthermore, after initiation, the two lateral organs have clear differences in their development. A conspicuous example is the presence of central vasculatures in LRs as opposed to peripheral vasculatures in nodules. Dissection of genetic pathways associated with the initiation of nodule and lateral root development has indicated a conservation between the primary root and LRs, but not nodules. Interestingly, nodules, have some similarities in cellular structures with that of the shoot [12].

Therefore, nodules, especially determinate nodules, differ from lateral roots on various properties, including the site of origin, type of initial cell divisions, meristem persistency, position relative to the parent cortex and vascular position [7]. The hormone requirements for development also appear to be different between these organs. For example, auxin is known to promote LRs [13,14], whereas there is very low auxin activity during nodule initiation. In fact, increased auxin activity has been shown to inhibit nodule formation, especially in determinate-nodule forming legumes [15,16]. On the other hand, cytokinin promotes nodule formation [17–19], but inhibits LR formation [17,20–22]. It is possible that initiation of LRs and nodules might be dictated by distinct auxin–cytokinin ratios. Similarly, not enough evidence is present to suggest if it originated from stem or carbon storage organs [7]. Therefore, whether nodules evolved by adopting the developmental signaling pathways of LR formation is an open question.

To address some of the outstanding questions on the similarities and differences in signaling and developmental pathways associated with the development of these lateral organs, we compared transcriptomes of lateral root and nodule tissues at two different stages of development in soybean. While a number of studies have evaluated gene expression during nodule formation [23–28], global gene expression profiles during LR formation has not been evaluated in legumes. In addition, the LR initiation process in legumes might be slightly different from that in *Arabidopsis* [9]. We employed RNA-seq to obtain transcriptome profiles, and used comparative analysis to identify the organ- and developmental stage-specific transcripts enriched in each organ.

## 2. Data Description

### 2.1. Overview of the Transcriptome Libraries

We dissected emerging nodules (EN, Figure S1A), mature nodules (MN, Figure S1B), emerging lateral roots (ELR, Figure S1C) and young lateral roots (YLR, Figure S1D), and constructed strand-specific RNA sequencing (RNAseq) libraries using polyA-enriched RNA preparations. Root sections above

and below these organs, devoid of any lateral organs (designated ABEN, ABMN, ABELR and ABYLR respectively), were used to construct respective control tissue libraries (Figure S1). We reasoned that "empty" root segments above and below the respective organs would serve as age-appropriate controls to identify organ-specific/enriched genes. At the early stage, both organs (EN and ELR) had minimal differentiation and had not completely emerged out of the primary root. We reasoned that comparison of transcriptomes at this stage would identify common developmental pathways—if any—that existed between initiation/formation of LRs and nodules. The mature stages were expected to identify distinct pathways that characterized specific functions of these organs. We harvested EN and ABEN from soybean (Williams 82) seedlings 8 days post-inoculation (dpi) with *Bradyrhizobium diazoefficiens* USDA110. MN and ABMN were harvested at 14–16 dpi. We distinguished EN as a slight bump in the root surface while mature MN were completely protruded and pink in color (Figure S1A,B). ELR, YLR, and their controls were harvested from 3–5 days old soybean seedlings not inoculated with *B. diazoefficiens*. ELR were selected based on a sharp protrusion with no tissue separation on the epidermis, which is also different from the bump seen at the cortical region of an emerging nodule. YLR were completely protruded and about 1–2 mm in length (Figure S1C,D). Three replicate sets of samples, each from independent experiments, were harvested for each of the eight tissue types for the preparation of 24 RNAseq libraries.

The libraries were sequenced using an Illumina HiSeq 2000, and RNAseq analysis was performed to identify organ-specific/enriched genes and pathways. Read mapping, transcript assembly and differential expression analyses were performed using the "Tuxedo" pipeline [29] (see experimental procedures for parameters). We used known nodule-specific genes as controls to verify the authenticity of the tissue-specific RNAseq libraries (Table 1). For example, *FRUIT WEIGHT 2.2 -LIKE 1* (*FWL1*) (Glyma09g31910) is induced early in the root hair and subsequently highly enriched in nodule tissues in response to rhizobium inoculation [28,30]. Consistently, we observed expression of FWL1 in both EN and MN; however, a significant enrichment was not observed in EN, likely due to its expression in adjacent control tissues colonized by rhizobia (Table 1). In contrast, we observed high expression and significant enrichment of *FWL1* in MN. *EARLY NODULIN GENE 40* (*ENOD40*) was enriched in both EN and MN, as expected. Similarly, all four symbiotic leghaemoglobins [31] tested were very highly expressed in MN, but their relative expression in EN tissues were less than 0.5% of what was observed in MN (Table 1). *NODULE INCEPTION* (*NIN*) and *NODULATION SIGNALING PATHWAY 1* (*NSP1*) also showed expected patterns of expression [32,33]. Importantly, the expression of these nodule marker genes was not detected in ELR, YLR or their controls, except in ENOD40, which is known to be expressed at low levels in the roots [34]. For lateral root markers, potential soybean orthologs of *Arabidopsis* genes enriched in LR tissues [35,36] were obtained from LegumeIP [37]. At least one ortholog of each family of lateral root primordium marker genes tested (*AUXIN RESPONSE FACTOR 5* (*ARF5*), *CYTOKININ RESPONSE FACTOR 2* (*CRF2*) and *LATERAL ROOT PRIMORDIUM 1* (*LRP1*)), except *GATA TRANSCRIPTION FACTOR 23* (*GATA23*), was significantly enriched in ELR and/or LR, as expected. Consistent with their role in LR formation, orthologs of *PIN-FORMED1* (*PIN1*) and *TARGET OF MONOPTEROS 7* (*TMO7*) were specifically enriched in ELR, and were highly expressed in LR tissues vs. nodule tissues. Importantly, none of the LR primordium markers were enriched in nodule tissues. Finally, a set of housekeeping genes, previously identified to be expressed uniformly in multiple soybean tissues [28], showed no difference in expression between the different organs and their control tissues (Table 1). Together, these results confirmed that our tissue harvests were of very high quality and specificity.

**Table 1.** Expression and enrichment of selected organ-specific genes.

| Transcript ID | Annotation | Expression Levels (FPKM) and Enrichment [1] | | | |
|---|---|---|---|---|---|
| | | EN | MN | ELR | YLR |
| *Nodule marker genes* | | | | | |
| Glyma09g31910.1 | FWL1 | 152.0 (0.0) | **1267.6 (8.0)** [c] | 0.0 (0.0) | 0.0 (0.0) |
| Glyma02g04180.1 | Enod40 | **1482.2 (2.9)** [c] | **1799.5 (4.7)** [c] | 12.4 (−1.0) | 13.2 (−2.3) |
| Glyma10g34290.1 | LBC_A | 0.2 (0.0) | **9191.6 (8.8)** [c] | 0.0 (0.0) | 0.0 (0.0) |
| Glyma10g34280.1 | LBC_C1 | 0.7 (0.0) | **9839.5 (9.0)** [c] | 0.0 (0.0) | 0.0 (0.0) |
| Glyma20g33290.1 | LBC_C2 | **27.9 (3.2)** | **6968.7 (9.0)** [c] | 0.0 (0.0) | 0.0 (0.0) |
| Glyma10g34260.1 | LBC_C3 | 19.3 (0.0) | **12680.1 (9.0)** [c] | 0.0 (0.0) | 0.0 (0.0) |
| Glyma04g00210.3 | NIN1 | 16.7 (0.0) | 84.8 (0.0) | 0.0 (0.0) | 0.2 (0.0) |
| Glyma16g01020.1 | NSP1 | **9.8 (3.5)** [c] | **19.7 (5.5)** [c] | 0.4 (0.0) | 0.7 (0.0) |
| *Lateral root marker genes* | | | | | |
| Glyma14g40540.1 | ARF5 | 6.7 (0.0) | 0.9 (−1.5) | 2.9 (0.0) | 2.6 (0.0) |
| Glyma17g37580.1 | ARF5 | 12.6 (0.0) | 2.5 (0.0) | **6.3 (1.0)** [a] | 6.3 (0.0) |
| Glyma05g37120.1 | CRF2 | 1.0 (0.0) | 2.2 (0.0) | **3.7 (2.0)** [c] | **1.6 (3.7)** [c] |
| Glyma08g02460.1 | CRF2 | 4.1 (0.0) | 7.2 (0.0) | **9.0 (1.9)** [c] | **6.1 (2.8)** [c] |
| Glyma03g39220.1 | GATA23 | 0.0 (0.0) | 0.1 (0.0) | 0.1 (0.0) | 0.0 (0.0) |
| Glyma19g41780.2 | GATA23 | 4.5 (0.0) | 6.3 (0.0) | 6.9 (0.0) | 5.0 (0.0) |
| Glyma02g44860.2 | LRP1 | 19.5 (0.0) | 8.6 (0.0) | 11.3 (0.0) | 6.1 (0.0) |
| Glyma07g35780.2 | LRP1 | 11.3 (0.0) | 6.0 (0.0) | 5.0 (0.0) | 3.3 (0.0) |
| Glyma14g03900.1 | LRP1 | 24.8 (0.0) | 9.8 (0.0) | **21.4 (0.7)** [a] | 9.4 (0.0) |
| Glyma07g11550.1 | PIN1 | 7.7 (0.0) | 2.2 (−2.6) | **24.7 (1.0)** [b] | 22.6 (0.0) |
| Glyma08g05900.1 | PIN1 | 8.2 (0.0) | 2.2 (0.0) | **19.6 (1.5)** [c] | 9.6 (0.0) |
| Glyma09g30700.1 | PIN1 | 7.8 (−1.0) | 1.7 (−3.2) | 15.5 (0.0) | 12.6 (0.0) |
| Glyma04g34080.1 | TMO7 | 0.0 (0.0) | 0.0 (0.0) | **5.2 (2.7)** [a] | **11.5 (5.0)** [a] |
| Glyma06g20400.1 | TMO7 | 0.2 (0.0) | 0.0 (0.0) | **23.9 (2.3)** [c] | 26.7 (0.0) |
| *Housekeeping genes* | | | | | |
| Glyma02g10170.1 | Actin11 | 149.7 (0.0) | 102.5 (0.0) | 204.5 (0.0) | 223.8 (0.0) |
| Glyma12g02310.1 | Cons4 | 14.1 (0.0) | 20.4 (0.0) | 17.9 (0.0) | 28.5 (0.0) |
| Glyma12g05510.1 | Cons6 | 24.2 (0.0) | 28.3 (0.0) | 36.6 (0.0) | 19.4 (0.0) |

[1] Expression levels are FPKM (fragments per kilobase of transcript per million mapped reads) values observed in each lateral organ tissue. Numbers in parenthesis indicate $\log_2$ fold change in expression level compared to the respective control tissues. Fold change values showing significant enrichment are highlighted in boldface and superscript letters indicate the level of statistical significance based on false discovery rate (FDR) adjusted p values ([a] $q < 0.05$; [b] $q < 0.01$; [c] $q < 0.001$).

### 2.2. Predominant Mapping to Coding Sequences and Consistency among Replicates in Our Libraries

We performed quality trimming (PHRED quality score >=20) and filtering (minimum read length = 25 nt) to improve read quality. This resulted in 28 million reads per library on average, which was expected to provide sufficient depth of coverage to reliably quantify gene expression using RNAseq [38]. The majority of the libraries retained ~85% reads post quality trimming and filtering; this indicated that these were high quality libraries. It should be noted that all three YLR libraries, all three ABYLR libraries, and one ELR library (replicate #1) had to be sequenced twice due to technical reasons and they retained only 65% reads, but 33 million reads on average (Table S1). We aligned the reads against the soybean reference genome (Gmax_v1.1_1.89; 54,175 gene models; 73,269 transcripts) allowing no mismatches or indels. On average, 84% of the high quality reads successfully mapped to the genome. Among these, ~18% were junction reads (that spanned two different exons; Table S1). Detailed examination of alignment positions in the genome demonstrated that most of the bases (76%) mapped to coding sequences, followed by the untranslated regions (UTRs) (18%). A smaller percentage of reads mapped to intronic (4%) and intergenic regions (2%) (Figure 1A). We compared the distribution of FPKM (fragments per kilobase of transcript per million mapped reads) values in each library (obtained from isoforms.read-group-tracking output of cuffdiff). Similar median values and FPKM distribution among the libraries (Figure S2) indicated that they could be reliably used to

compare gene expression. We performed hierarchical clustering based on correlation distances and the results indicated excellent consistency among the three different replicates of the same tissue types (Figure 1B). Together, these data indicated that our libraries were of very high quality and very well suited for global gene expression analysis. In Table S2, a master gene expression matrix of transcript abundance in each replicate sample, calculated enrichment in each tissue type, and corresponding false discovery rate (FDR)-adjusted *p* values are presented.

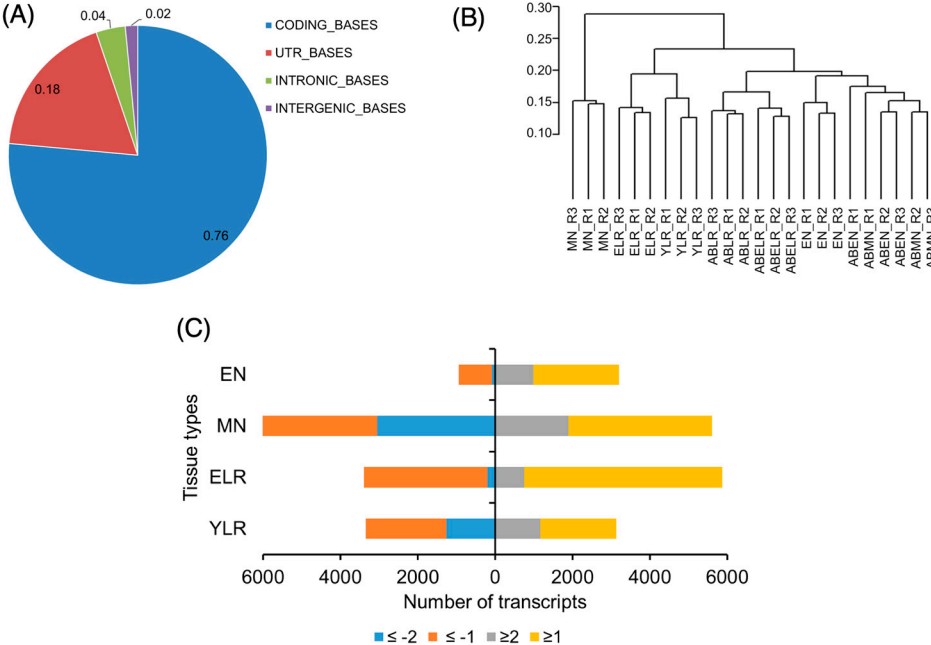

**Figure 1.** (**A**) Post alignment summary of reads mapped to the soybean reference genome from all 24 libraries. The proportion of nucleotide bases mapping to coding regions (blue), untranslated regions (UTRs) (red), introns (green) and intergenic regions (purple) are indicated; (**B**) a dendrogram resulting from hierarchical clustering of transcript fragments per kilobase of transcript per million mapped reads (FPKM) values using correlation distances between the different libraries; (**C**) numbers of significantly differentially-expressed transcripts in each tissue. Only those with FPKM ≥1 in at least one tissue type are shown. Proportion of transcripts with log$_2$ fold change of ≤−1 (orange), ≤−2 (blue), ≥1 (grey), and ≥2 (yellow) in the emerging nodule (EN), mature nodule (MN), emerging lateral root (ELR) or young lateral root (YLR) vs. the respective control tissues are indicated.

### 2.3. Mature Nodules Had the Largest Difference in Global Gene Expression Patterns

We compared the global transcriptome in each lateral organ to its corresponding control to identify organ-enriched transcripts (Table S3). A minimum expression threshold of FPKM ≥1 was used to reliably identify transcripts differentially expressed in each organ versus its control tissue (EN vs ABEN, MN vs ABMN, ELR vs ABELR and YLR vs ABYLR). Those that were differentially expressed were further classified based on the extent of differential expression, i.e., log$_2$ significant fold change of ≥1, ≥2, ≤−1 and ≤−2. MN and ELR had the largest number of transcripts differentially expressed relative to their control root segments (Figure 1C). Among all four lateral organ tissues, MN had the highest number of transcripts that were down-regulated compared to ABMN. Overall, MN had the largest number of genes differentially expressed vs. its control tissue. This can be attributed to specialization of MN in both the metabolic and developmental pathways relative to root tissues. At younger stages of both these lateral organs (EN and ELR), there were more transcripts that were up-regulated compared to ABEN and ABELR than the ones that were down-regulated (Figure 1C), suggesting that a number of developmental pathways are being activated in these tissues.

Global comparative analysis of enriched biological processes in each tissue type (Table S4) not only confirmed expected enrichment patterns (e.g., purine, organic acid, and adenosine triphosphate [ATP] synthesis in mature nodules), but also revealed new biological insights. For example, DNA modification (methylation and alkylation) were specifically enriched in EN tissues, suggesting epigenetic regulation during nodule development. Enrichment of arginine metabolism and iron transport provided strong additional evidence for nitric oxide regulation during LR development. Both EN and ELR were enriched in cell division and associated gene sets during early stages, but distinguish themselves in general at the transcriptome level, consistent with the distinct identities of these organs at later stages.

## 3. Methods

### 3.1. Plant Material and RNA Isolation

Soybean (*Glycine max*) cv. Williams 82 seeds were surface-sterilized [39] and sown in a mixture of autoclaved vermiculite and perlite (2:1 ratio). Seedlings were grown in a growth chamber (Conviron, Manitoba, Canada) at 25 °C with a 16/8 hrs light cycle, and watered with nitrogen-free plant nutrient solution (N- PNS; [39] for both nodule and root tissue harvest. To harvest emerging (EN) and mature (MN) nodules and their controls, plants were inoculated with *B. diazoefficiens* USDA 110 (grown in Vincent rich medium [40] at 30 °C shaking at 200 rpm). For inoculation the *B. diazoefficiens* were resuspended in N- PNS to a concentration of 0.08 OD at 600 nm [41] and applied to the roots. EN, MN and control tissues were harvested at 5–7 dpi and 14–16 dpi respectively. ELR and YLR and their controls were harvested from uninoculated plants. All growth conditions, including watering, were identical between plants used for LR and nodule harvests except rhizobium inoculation. ELR, YLR tissues and controls were harvested 5–7 days after germination. For lateral organ harvests, plants were removed from vermiculite and perlite mixture, washed thoroughly in sterile water to remove the dirt, and organs were dissected using sterile scalpel blades under a dissection microscope. For each replicate, harvests were collected from at least 15–20 plants and pooled. The dissected tissues were wipe dried on a sterile paper towel and collected in TRI reagent (Sigma Aldrich, St. Louis, MO, USA) in a pre-weighed 2 mL microcentrifuge tube on ice, and stored at −80 °C until the RNA was isolated. Plant growth, inoculations and tissue harvests were performed in batches for each of the three biological replicates. Total RNA was isolated using the protocol recommended by the manufacturer with slight modifications [39]. The major exception was that a tissue lyser (QIAGEN, Hilden, Germany) was used for tissue homogenization rather than manual grinding with a pestle and mortar.

### 3.2. Transcriptome Library Preparation and Sequencing

We prepared directional RNAseq libraries from each of the 24 RNA preparations using 5 μg of total RNA for each library (4 lateral organ tissues + 4 respective control tissues) × 3 replicate harvests). The RNA quality was checked using a Bioanalyzer (Agilent Technologies, Santa Clara, CA, USA), and each library was prepared using a ScriptSeq™ v2 RNA-Seq Library Preparation Kit (Epicentre, Madison, WI, USA) in accordance with the manufacturer's recommendation. Briefly, PolyA RNA was isolated, and reverse transcribed using random hexamer primers with a 5′ tagging sequence. The 5′ tagged cDNAs were then tagged at their 3′ ends using terminal tagging oligos (TTOs) with blocked 3′ ends. The resulting di-tagged cDNA was linearly amplified using primers with adapter sequences [42]. The amplified libraries were then sequenced on an Illumina HIseq2000 (single end, 50 nt read length) using four lanes and six samples per lane. The three replicates of each lateral organ tissue and its control were run on a single lane. At the end of the run, sequences were multiplexed into 24 different sequences files using the specific tag barcodes. It should be noted that all three YLR and ABYLR libraries, as well as one ELR library (replicate #1), had to be sequenced twice due to technical reasons. The outputs from both runs were merged to obtain a sufficient number of high quality reads. All library construction, sequencing, and adapter trimming were performed at the Genomics Core Facility, University of Missouri, Columbia, MO, USA.

*3.3. Quality Control of Raw Reads*

The average sequencing yield was 36 million reads per library (Table S1). The read quality was evaluated using *fastqc* -v0.10.1 (http://www.bioinformatics.babraham.ac.uk/projects/fastqc/). The results suggested that a portion of the reads might have few nucleotides at the 5′ end and ~7–10 nts at the 3′ end with a poor PHRED score (<20). The sequences were filtered and trimmed for quality using *prinseq-lite* -v0.19.5 [43]. We trimmed the reads at both the ends using the following parameters: Minimum PHRED quality score = 20, window size of 5, no Ns, and minimum post trimming length = 25. After trimming, the read quality was again checked using FastQC, which indicated significant improvement in quality with the all nucleotides with PHRED quality score >20.

*3.4. Read Alignment and Assembly*

RNAseq read alignment and assembly was carried out based on the "Tuxedo" pipeline [29]. In brief, reads were aligned using *tophat* (v2.0.5) to the soybean reference genome (Gmax_v1.1_189, Phytozome v9.0). For differential expression analysis, each library was mapped individually, and for analysis of overall mapping statistics and coverage vs. FPKM analysis, all the libraries were mapped together. *tophat* was run with the following parameters: No transcriptome, genome read, no read and segment mismatches, no deletions and insertions, maximum intron length -5000, and library type-fr-second strand. The mapping was guided using gene models in Gmax_v1.1_189_gene.gff3 and was run without the –no-novel junctions option to identify the novel splice junctions. The mapped reads from each library were assembled into transcripts using *cufflinks*, where the transcriptome file was used as a guide and the genome sequence file was used to enhance assembly [29]. The maximum intron length was kept at 5000, library type was fr-second strand and u-option was used to accurately weigh multiple mappings of the same read. The resulting assemblies were then merged using *cuffmerge* using both the transcriptome and genome file as guide to obtain a master gtf reference file, which combined the novel splice junctions identified from this study with previously annotated ones. Differential expression analysis between the different lateral organs vs. respective controls was performed using *cuffdiff* (FDR < 0.05). Note that all of these three tools, *cufflinks*, *cuffmerge* and *cuffdiff*, are available in the suite cufflinks (*cufflinks* v2.0.2). Instead of directly using the fold change output provided by *cuffdiff*, we calculated "significant fold change", where the fold change was converted to zero if the stat test was not significant. This helped to filter out transcripts with a fold change, but the statistics test was either invalid or the result was not significant. Post-alignment summaries were obtained by using *samtools* -v0.1.18 [44] and *picard* -v1.98 (http://broadinstitute.github.io/picard/). *Samtools* was used to obtain the number of mapped reads and unique reads for each library. The CollectAlignmentSummaryMetrics tool in *picard* was used to obtain the mapping quality and genomic position annotation for each mapped base.

*3.5. Singular Enrichment Analysis*

For a global comparison of biological processes among lateral organs, transcripts with FPKM $\geq 1$ and significant $\log_2$ fold change of $\geq 1$ (compared to their controls) were used as an input for singular enrichment analyses using AgriGO-1.2 [45]. The analysis was performed using soybean genome locus IDs (Phytozome v1.1) and a complete set of gene ontologies. Biological processes that were significantly enriched (Fisher statistical test method with Yekutieli (FDR under dependency), significance level <0.05) were selected and listed in Table S4.

## 4. User Notes

The large-scale transcriptomics data set generated in this study will serve as an excellent resource for discoveries by the community. Example uses of the dataset include but are not limited to (i) the generation of global gene regulatory networks using co-expression analysis [46]; (ii) building transcription factor-specific or lateral organ-specific networks using graphical Gaussian

models [47]; (iii) metabolic pathway analyses of carbon and nitrogen metabolism, specialized secondary metabolites, lipid biosynthesis and metabolism which have been suggested to play a role in organ development [48–50]; (iv) comparative analyses of key gene families, e.g., transporters, P450 monooxygenases, Mitogen Associated Protein Kinase (MAPK) cascade components and resistance (R) genes [51–53]; (v) systems biology analyses combined with proteomics and metabolomics datasets [54,55]; and (vi) organ-specific alternate spliced transcripts [56]. When combined with other similar datasets, especially from leguminous plants, these analyses can help answer questions on the evolutionary origins of root nodules and relationships between the development of different plant lateral organs.

**Supplementary Materials:** The following are available online at http://www.mdpi.com/2306-5729/4/2/64/s1, Figure S1: Representative images of root lateral organs and corresponding control tissues harvested for transcriptome library construction, Figure S2: Box plots showing the distribution of FPKM values for all 24 libraries, Table S1: Read quality and mapping summary of the 24 RNA Seq libraries analyzed in this study, Table S2: Global transcriptome changes in each lateral organ tissue type, Table S3: List of tissue-enriched transcripts, Table S4: Summary of biological processes enriched in each tissue type.

**Author Contributions:** Conceptualization, S.S.; methodology, S.A. and S.D.; formal analysis, S.A. and S.D.; data curation, S.A. and S.S.; writing—original draft preparation, S.A.; writing—review and editing, S.D. and S.S.; visualization, S.A.; supervision, S.S.; project administration, S.A. and S.S.; funding acquisition, S.S.

**Funding:** This research was supported by funds from South Dakota State University, SD Agricultural Experiment Station (H351-09 and H543-15), SD Soybean Research and Promotion Council (SA1400515), the National Science Foundation/EPSCoR Cooperative Agreement #IIA-1355423, and the State of South Dakota.

**Acknowledgments:** Technical assistance provided by the DNA Core facility, University of Missouri in library construction and high throughput sequencing, the use of South Dakota State University's high performance computing clusters for data analysis, technical support from South Dakota State University's research information technology team (Brian Moore and Alan Carter) are gratefully acknowledged.

**Conflicts of Interest:** The authors declare no conflict of interest. The funders had no role in the design of the study; in the collection, analyses, or interpretation of data; in the writing of the manuscript, or in the decision to publish the results.

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
