# Peer review of "Lateral Root and Nodule Transcriptomes of Soybean"

_data, 2019_

Reviewer 1 Report

The manuscript "Lateral root and nodule transcriptomes of soybean" by Adhikari et al., assessed for publication MDPI Data. This is a simple straight forward comparative transcriptomics study, which has generated a vast amount of valuable expression data (resource). The authors have performed RNAseq on four samples emerging nodules, mature nodules, emerging lateral roots and young lateral roots, and assessed the quality of the data and cross checked a handful of genes which was reported (marker genes including the house keeping genes) to be differentially or specifically expressed on the experimental conditions. The manuscript is well written, although the analysis was shallow. As this data is expected to serve as a resource for the community the quality of the data is more important at this stage, rather than the indepth analysis. Only few corrections,

Line 57 – suggested that indicated

Table 1: It will be useful to include pvalue in the expression table.

Author Response

The manuscript "Lateral root and nodule transcriptomes of soybean" by Adhikari et al., assessed for publication MDPI Data. This is a simple straight forward comparative transcriptomics study, which has generated a vast amount of valuable expression data (resource). The authors have performed RNAseq on four samples emerging nodules, mature nodules, emerging lateral roots and young lateral roots, and assessed the quality of the data and cross checked a handful of genes which was reported (marker genes including the house keeping genes) to be differentially or specifically expressed on the experimental conditions. The manuscript is well written, although the analysis was shallow. As this data is expected to serve as a resource for the community the quality of the data is more important at this stage, rather than the indepth analysis.

            Thank you for your time and detailed assessment of the manuscript.

Only few corrections,

Line 57 – suggested that indicated

            Thanks. This has been corrected.

Table 1: It will be useful to include pvalue in the expression table.

            Thanks. We have indicated the FDR corrected p-values using superscript letters in Table 1.

Reviewer 2 Report

- Comments to the Authors

This paper by Adhikari et al. provides high quality data to compare transcriptomes of soybean  from lateral roots and from nodules. These data have interesting potential uses, but some changes can be considered in the current manuscript

The text (data description and Table 1) should mention new possible gene regulatory networks or key gene families identified in this work associated specifically to emerging or mature nodules or to emerging or young lateral roots.

Minor points

- Check sentence on line 57/58.

- A reference should be included at the end of sentence on line 57/58

- Footnote of Fig S1  have errors corresponding to panels C and D

- Line 110, B. diazoefficiens USDA110

- Lines 112 and 116, Figs S1A, B, C or D

- Line 135, At least

- Table 1, indicate p value

- Lines 189 and 195; Figure 1d does not exist

- Line 247 “Tuxedo” as in line 120

- Line 263, include a reference following (cufflinks v2.0.2)

Author Response

This paper by Adhikari et al. provides high quality data to compare transcriptomes of soybean  from lateral roots and from nodules. These data have interesting potential uses, but some changes can be considered in the current manuscript

            Thank you for your time and the assessment that the manuscript provides high quality data

The text (data description and Table 1) should mention new possible gene regulatory networks or key gene families identified in this work associated specifically to emerging or mature nodules or to emerging or young lateral roots.

We agree with the reviewer that new gene regulatory networks and key gene families will be a useful addition. However, we believe that these analyses will be outside the scope of a resource paper. We have instead included a list of biological processes specifically enriched in each of the four tissues as well as those enriched in multiple tissues (Table S4), and discussed this information in the text[SS1]  highlighting what is new.

Minor points

- Check sentence on line 57/58.

            Thanks. This has been corrected.

- A reference should be included at the end of sentence on line 57/58
            Thanks. A reference is now included.

- Footnote of Fig S1 have errors corresponding to panels C and D
            Thanks for catching this error. We have corrected it now.

            - Line 110, B. diazoefficiens USDA110

Thanks. The strain information is now included.

- Lines 112 and 116, Figs S1A, B, C or D
            Thanks. This has been corrected.

- Line 135, At least
            This sentence has been modified for clarity in the revised version.

- Table 1, indicate p value

Thanks. We have now indicated the FDR corrected p-values using superscript letters in Table 1.

- Lines 189 and 195; Figure 1d does not exist

Thanks. This has been corrected.

- Line 247 “Tuxedo” as in line 120
            Thanks. This has been corrected.

- Line 263, include a reference following (cufflinks v2.0.2)
            Thanks. A reference is now added.

Reviewer 3 Report

The manuscript by Adhikari et al. describes a new transcriptome dataset that compares lateral roots and nodules in soybean. The manuscript is original and the data generated will add to the existing transcriptomic resources. The manuscript reads well and the experimental approach is described in sufficient details. The comparison between nodules/LRs and adjacent root segments provides a good control dataset. In addition, testing transcript levels of nodule and root-specific markers provided good control for data quality and specificity.

Suggestions to improve the manuscript quality:

-       The first couple of sentences in the paragraph starting at line 57 need to be reformulated for clarity

-       In Fig. S1, panel D seems to show a young lateral root, but the legend does not include this information. Please reformulate the legend to clearly describe what each panels shows. From this figure it is also unclear which root regions are the controls

-       Do the three replicate samples come from tissue harvested in independent experiments, or do all samples come from one single experiment (biological replicate)? This should be clarified.  If all samples were harvested from plants grown in one single experiment, the presented data may not be reliable. What is the sample size for each biological replicate (from how many plants were root/LR samples harvested)? If the biological replicate represented 4 lateral root or 4 nodules, the sample size is very low and may not yield reliable data.

-       To improve the value of the presented dataset, in addition to gene IDs, gene annotations should be included in Tables S2 and S3.

Due to my concerns regarding sample size and biological replicates, it is difficult to judge the overall data quality.

The data was deposited in the NCBI GEO and SRA. An accession number was assigned.

The scientific community would benefit more if the data is integrated in the soybean database (soybase.org).

Author Response

The manuscript by Adhikari et al. describes a new transcriptome dataset that compares lateral roots and nodules in soybean. The manuscript is original and the data generated will add to the existing transcriptomic resources. The manuscript reads well and the experimental approach is described in sufficient details. The comparison between nodules/LRs and adjacent root segments provides a good control dataset. In addition, testing transcript levels of nodule and root-specific markers provided good control for data quality and specificity.

            Thank you for your detailed assessment of the manuscript and agreeing that the data are of good quality.

Suggestions to improve the manuscript quality:

-       The first couple of sentences in the paragraph starting at line 57 need to be reformulated for clarity

            Thanks. This has been corrected.

-       In Fig. S1, panel D seems to show a young lateral root, but the legend does not include this information. Please reformulate the legend to clearly describe what each panels shows. From this figure it is also unclear which root regions are the controls

Thanks for catching this. This has been corrected and the figure legend revised for clarity.

-       Do the three replicate samples come from tissue harvested in independent experiments, or do all samples come from one single experiment (biological replicate)? This should be clarified.  If all samples were harvested from plants grown in one single experiment, the presented data may not be reliable.

Yes, The three replicate samples came from independent tissue harvests from separate experiments (This information has now been included in the manuscript). Each replicate sample was collected from multiple plants under a dissection microscope to obtain clean nodule/lateral root samples and control tissues adjacent to these organs i.e. root segments above and below without any nodule and lateral root.

- What is the sample size for each biological replicate (from how many plants were root/LR samples harvested)? If the biological replicate represented 4 lateral root or 4 nodules, the sample size is very low and may not yield reliable data.

-       Each replicate tissue was collected from at least 15-20 plants and pooled. This information has been added to the manuscript text.

- To improve the value of the presented dataset, in addition to gene IDs, gene annotations should be included in Tables S2 and S3.

            Thanks for the suggestion. This information has been added now. 

- Due to my concerns regarding sample size and biological replicates, it is difficult to judge the overall data quality.

We have clarified the sample size and biological replicates information.

The data was deposited in the NCBI GEO and SRA. An accession number was assigned.

The scientific community would benefit more if the data is integrated in the soybean database (soybase.org). 

Thanks for the suggestion. We have initiated contact for integrating the dataset into soybase.

Data EISSN 2306-5729 Published by MDPI AG, Basel, Switzerland RSS E-Mail Table of Contents Alert
Back to Top